# The Role of Cytoskeleton Protein 4.1 in Immunotherapy

**DOI:** 10.3390/ijms24043777

**Published:** 2023-02-14

**Authors:** Chaohua Si, Lihua Yuan, Chen Chen, Ting Wang, Qiaozhen Kang

**Affiliations:** School of Life Science, Zhengzhou University, Zhengzhou 450001, China

**Keywords:** cytoskeleton protein 4.1, tumor suppressor, immunotherapy, tumor microenvironment, cancer

## Abstract

Cytoskeleton protein 4.1 is an essential class of skeletal membrane protein, initially found in red blood cells, and can be classified into four types: 4.1R (red blood cell type), 4.1N (neuronal type), 4.1G (general type), and 4.1B (brain type). As research progressed, it was discovered that cytoskeleton protein 4.1 plays a vital role in cancer as a tumor suppressor. Many studies have also demonstrated that cytoskeleton protein 4.1 acts as a diagnostic and prognostic biomarker for tumors. Moreover, with the rise of immunotherapy, the tumor microenvironment as a treatment target in cancer has attracted great interest. Increasing evidence has shown the immunoregulatory potential of cytoskeleton protein 4.1 in the tumor microenvironment and treatment. In this review, we discuss the role of cytoskeleton protein 4.1 within the tumor microenvironment in immunoregulation and cancer development, with the intention of providing a new approach and new ideas for future cancer diagnosis and treatment.

## 1. Introduction

Cancer has always been a great challenge for humans, and researchers are constantly innovating and improving treatment technologies [1]. Immunotherapy produces significantly fewer side effects than the very common chemoradiotherapy, and it has been shown to be effective in some patients [2]. Complete or long-acting responses are higher with immunotherapy compared to conventional therapies, but there is still room for improvement, as there are still many patients who are or have become primary resistant to immunotherapy [3]. Immunotherapy, which involves activating the body’s immune system to kill cancer cells, has shown tremendous anti-tumor activity in various cancers, including melanoma, kidney, lung, and prostate cancer [4]. The success of immunotherapy has benefited many patients, but there is still a subset of patients who are insensitive to immunotherapy, in part due to suppressive responses within the tumor microenvironment [5]. With an increasing understanding of immunotherapy, we have come to realize that the tumor microenvironment’s homeostasis is crucial to immunotherapy’s efficacy. Undoubtedly, a deeper understanding of the role of the tumor microenvironment in immunotherapy is essential to improve efficacy.

In studying the tumor microenvironment and immunotherapy, we noticed cytoskeleton protein 4.1, which has four types: 4.1R, 4.1N, 4.1G, and 4.1B [6]. They act in a synergistic way in tumor cells or immune cells by affecting cell structure, motility, and most other processes necessary for the survival and development of normal and cancer cells. Cytoskeleton protein 4.1 can often regulate the progression of cancer by inhibiting the growth and survival of tumor cells during the malignant transformation of cells [7]. Tumor cells can utilize intracellularly related signaling pathways to promote their malignant features, while cytoskeleton protein 4.1 often inhibits cancer progression via interaction with multiple signaling pathways [8,9].

In this article, we present a review on the role of cytoskeleton protein 4.1 in the tumor microenvironment and immunotherapy. We elucidate how this protein affects the function of tumor cells and immune cells, with the aim of providing new ways and new ideas for the diagnosis and treatment of cancer.

## 2. Immunoregulation and Tumor Microenvironment

Immunomodulation refers to the body’s reliance on the immune system to recognize and remove foreign antigenic substances, maintain the body’s dynamic physiological balance, and keep the body’s immune response at the most appropriate level [10]. The malignant metastasis of tumors is an important reason for cancer patients’ high mortality rate [11,12]. There are three ways of tumor metastasis: direct metastasis, lymphatic metastasis, and hematologic metastasis. Tumor metastasis is a complex process, which requires a series of molecular changes to promote the progression of metastasis, including enhanced invasiveness and migration [13]. Cytoskeleton protein 4.1 is able to maintain cell adhesion and protein cytoskeleton organization, and is involved in mediating processes such as cell growth and motility. The metastasis of tumor cells often starts in the tumor microenvironment and is regulated by the immune system [14].

Depending on the number and spatial distribution of T cells, the tumor microenvironment can be divided into three phenotypes: immune inflammatory, immune rejection, and immune desert, and thus immunotherapy has different effects on different phenotypes [15]. First, the immune inflammatory type refers to the presence of T cells located in the parenchyma and capable of expressing pro-inflammatory cytokines [16]. Secondly, the immune rejection type refers to the presence of immune cells around the tumor or mechanisms that are unable to enter the parenchyma because they are obstructed by the vascular stroma [16]. In contrast, the immune desert type lacks antitumor immune cells, and this phenotype is least responsive to immune checkpoint inhibitors [16,17].

For immunotherapy to successfully inhibit cancer progression, three steps must be accomplished: successful activation of the immune system, successful expansion of effector cells, and successful infiltration of effector cells into tumor tissue and the destruction of tumor cells. However, some patients do not benefit from this because the tumor microenvironment can limit drug delivery [18]. Even if the drug can be delivered to the target site, the tumor microenvironment can affect its efficacy [19]. The tumor microenvironment can also respond to immunotherapy by affecting lymphocyte sensitization, reducing effector cell expansion, and inhibiting the ability of effector cells to infiltrate [20]. The tumor microenvironment is a suppressive microenvironment that creates a biological barrier that envelops tumor tissue. It attenuates the efficacy of immunotherapy by depleting specific T cell populations and attenuating the sensitization of T cells [21]. Fortunately, the tumor microenvironment is dynamic [22]. The dynamic regulation and interactions of various immune cells therein can also significantly influence the malignant properties of tumor cells. Early detection, diagnosis, and treatment will significantly improve cancer patients’ survival rate and quality of life [23]. If we can detect and diagnose in time and take some effective measures, such as improving the body’s immunity, it will significantly improve the treatment effect of patients (Figure 1).

## 3. Cytoskeleton Protein 4.1

Cytoskeleton protein 4.1 members are encoded by different paralogous homologs, but all of its family members share three highly conserved structural and functional domains, namely, the actin-binding (SAB) and c-terminal (CT) domains, and a highly conserved structural domain consisting of 200–300 amino acids [7]. Through the three highly conserved structural domains, Cytoskeleton protein 4.1 is able to bind to membrane molecules, which is the basis for the role of Cytoskeleton protein 4.1 in signal transduction. In addition, Cytoskeleton proteins have three unique structural domains, U1, U2 and U3, and they confer different characteristics to the Cytoskeleton protein 4.1 family members [6,24,25] (Figure 2A). Cytoskeleton protein 4.1 was initially found to act as an adapter capable of connecting membrane proteins to cytoskeleton proteins to the cytoskeleton [26]. It is an essential component of the membrane backbone and plays a key role in membrane backbone stability and protein complex assembly. As research progressed, researchers found that Cytoskeleton protein 4.1 not only exists as an adapter and membrane skeleton, but also directly participates and influences various cellular activities such as migration [27,28]. Uncontrollable metastasis is one of the most remarkable biological features of malignant tumors [29]. Cytoskeleton protein 4.1 is localized on the cell membrane, and in addition to acting as a cytoskeleton to maintain cell morphology, it is also able to influence the migration of tumor cells or other malignant features such as proliferation and signal transduction through multiple signaling pathways and molecules [27] (Figure 2B–E).

## 4. The Role of Cytoskeleton Protein 4.1 in Immunoregulation and Cancer

Currently, depending on the type of cancer and the severity of cancer, patients are treated by surgery, chemotherapy, radiotherapy, immunotherapy, or other means [30,31]. In contrast to surgery and radiotherapy, immunotherapy influences cancer progression by modulating the immune system and immune cells to inhibit the metastasis and proliferation of tumor cells [32]. The presence of a large number of tumor cells and immune cells within the tumor microenvironment can greatly influence the effectiveness of immunotherapy [33]. Many studies have shown that cytoskeleton protein 4.1 deficiency is closely related to the progression of cancer development. Promoting high expression of cytoskeleton protein 4.1 can improve the effect of immunotherapy to some extent [34,35]. In addition, the study results show that cytoskeleton protein 4.1 can affect both the function of immune cells and the malignant characteristics of tumor cells, making it a potential target for cancer diagnosis and treatment. In the following, we provide an overview of how cytoskeleton protein 4.1 in the tumor microenvironment affects immune cells and tumor cells (Figure 3).

For example, B cells are specific antigen-presenting cells usually involved in humoral immunity. B cells exert their immune response through class switch recombination (CSR) and plasma cell differentiation (PCD). The activation process of B cells is associated with membrane-bound proteins and adapters, while cytoskeleton protein 4.1 can attach membrane proteins to the cytoskeleton and participate in processes such as cell differentiation with adaptor properties [36]. T cells receptor-mediated signaling pathways are critical for CD8+ T cell function, and cytoskeleton protein 4.1 can inhibit CD8+ T cell activation by inhibiting LAT phosphorylation [37]. Macrophages are an essential component of human nonspecific immunity and a vital cell population in the tumor microenvironment, because they play a role in regulating chemotaxis, phagocytosis, and inflammatory responses, while cytoskeleton protein 4.1 can regulate tumor progression by affecting vascular endothelial growth factor A (VEGFA) expression [38]. Dendritic cells are antigen-presenting cells that play a crucial role in thymic immunity and autoimmune diseases. Their migration is often associated with protective inflammatory responses and allergic immune responses, and cytoskeleton protein 4.1 inhibits cell cycle arrest and influences the progression of cancer development by affecting the migratory capacity of dendritic cells [39]. In contrast to acting directly on the immune cells themselves, cytoskeleton protein 4.1 often acts on their growth factors to influence cancer development and progression in tumor cells. For example, in colon cancer, cytoskeleton protein 4.1 regulates the secretion and expression of VEGFA, which affects the proliferation, migration, angiogenesis, and invasion of colon cancer cells [40]. Similarly, cytoskeleton protein 4.1 can act synergistically with the cell surface receptor amyloid precursor protein to inhibit the migration and invasion of kidney cancer cells. In addition, cytoskeleton protein 4.1 is also present in melanoma, breast cancer, and squamous cell lung cancer [41].

## 5. Function of Cytoskeleton Protein 4.1

Cytoskeleton protein 4.1 is an important tumor suppressor molecule. It is expressed, downregulated, or absent in various cancers, which in turn affects tumor cell proliferation and migration, and others to inhibit cancer development and progression. The tumor microenvironment is the site of tumor cell genesis and development, and most of the effects of cytoskeleton protein 4.1 on tumor cells occur in the tumor microenvironment [42]. Existing studies suggest that cytoskeleton protein 4.1G usually acts as an adapter between transmembrane proteins and the cytoskeleton, and its function in cancer is still largely unknown. Therefore, we only reviewed the mechanisms by which other members of cytoskeleton protein 4.1 affect cancer in the tumor microenvironment [27,43,44] (Figure 4).

### 5.1. Protein 4.1R

Meningiomas are common central nervous system tumors, and an NF2 inactivation is an early event in the development and progression of meningiomas [45]. A study by Robb et al. showed that protein 4.1R is frequently expressed at reduced levels or is even absent in meningioma patient tissues compared to normal subjects, and the same results were found in two meningioma cell lines, IOMM-Lee and CH157-MN [46]. The *NF2* gene encodes Merlin protein, which is structurally related to protein 4.1R, is enriched to the membrane during growth arrest, and interacts with a subset of Merlin proteins associated with tumor suppressor functions and components such as CD44 to inhibit the proliferation of meningioma cells and negatively regulate cell growth. M2 macrophages are an important class of immune cells within the tumor microenvironment [47]. In colon cancer, protein 4.1R expression was downregulated. Lu et al. showed that the knockdown of protein 4.1R reduced the phagocytic capacity of M2 macrophages and affected the immune process. The mechanism was that knockdown of protein 4.1R upregulated VEGFA secreted by M2 macrophages and activated the PI3K/AKT pathway, which inhibited the immune effect of M2 macrophages and promoted the proliferation, migration, invasion, etc., affecting the process of colon cancer development and progression, and is a potential therapeutic target for colon cancer [38]. Small cell lung cancer (SCLC) is a group of neuroendocrine tumors with rapid disease progression and a short natural course, most often metastatic at diagnosis, and usually exhibits chemoresistance, yet treatment with immune checkpoint inhibitors significantly improves overall patient survival [48]. Cell Adhesion Molecule 1 (CADM1) has been reported to influence the development of small cell lung cancer through its ability to inhibit the proliferation and metastasis of the disease. At the same time, CADM1 cytoplasmic structural domain mutants are excellent diagnostic markers and therapeutic targets for small cell lung cancer. Funaki et al. showed that protein 4.1R is a CADM1 downstream binding site in non-small cell lung cancer; that CADM1 can recruit protein 4.1R to the plasma membrane, promoting anchorage-independent growth of small cell lung cancer (SCLC) cells and influencing the development of small cell lung cancer; and that co-localization of the CADM1 protein 4.1R complex at the plasma membrane correlates with the pathological stage of small cell lung cancer and reflects the malignancy of small cell lung cancer pairs [49].

### 5.2. Protein 4.1N

Ovarian cancer is the leading cause of death from gynecological cancers worldwide, with epithelial ovarian cancer accounting for 90% of ovarian cancer patients, and peritoneal spread being the primary process of metastasis in epithelial cancer [50]. Wang validated the effect of protein 4.1N on epithelial ovarian cancer (EOC) patients in two EOC cell lines: SKOV3, which endogenously expresses 4.1N, and A2780, which lacks 4.1N expression. The results showed that protein 4.1N-silenced wild-type A2780 cells (A2780 WT) caused downregulation of epithelial-mesenchymal transition (EMT) markers such as N-calmodulin expression. It also affected the migration and invasion of EOC cells and promoted their susceptibility to loss-of-nest apoptosis. Thus, the study by Wang et al. confirmed that protein 4.1N deficiency in ECO patients promotes the process of EMT. Furthermore, protein 4.1N inhibited nest loss resistance and EMT progression in EOC cells by promoting its degradation through direct binding to 14-3-3 [51]. Breast cancer is one of the most common cancers in women, and the ease of metastasis is the main reason for its high mortality rate [52]. Ji et al. showed that protein 4.1N is differentially expressed in cell lines with different metastatic abilities and is not expressed in breast cancer cell lines with high metastatic properties, revealing the vital role that protein 4.1N plays in breast cancer metastasis and revealing that protein 4.1N is a negative regulator affecting cancer cell adhesion, migration, and invasion [53]. Wang et al. showed that the expression level of protein 4.1N was negatively correlated with the metastatic ability of small cell lung cancer cell lines and that low expression of protein 4.1N promoted the progression of non-small cell lung cancer [54]. The FERM domain of protein 4.1N, which is required for its localization and stabilization in the basement membrane, binds to protein phosphatase1 (PP1), and PP1-mediated dephosphorylation contributes to the localization of protein 4.1N. Activation of the JNK-c-Jun pathway is pro-cancerous in lung cancer, and PP1 inactivates JNK, thereby preventing cancer progression. Protein 4.1N can recruit PP1 to affect the processes of adhesion and proliferation in non-small cell lung cancer by inhibiting critical molecules of the JNK-c-Jun pathway. In microcarcinoma, the CTD structural domain of protein 4.1N can bind to the N-terminal end of Pike, an enhancer of PI3K, and the competitive binding effect of the two leads to cell cycle alterations, which in turn affects the progression of cancer [54].

### 5.3. Protein 4.1B

Gastric cancer is one of the most common malignancies in China and has a high incidence in the country, but the mechanisms regulating it are still very unclear [55]. The surface epidermal growth factor receptor (EGFR) plays an essential role in regulating metabolism, growth, and differentiation, and it has a potent mitogenic activity that can stimulate or inhibit tumor cell growth [56]. Xue et al. showed that protein 4.1B could bind EGFR and inhibit EGFR function through the EGFR/MAPK/ERK1/2 pathway, thereby regulating the development and progression of gastric cancer [57]. Protein 4.1B has been shown to inhibit cancer in various cancers and is partially regulated by inducing apoptosis in tumor cells [58]. In breast cancer, DAL-1 (Differentially Expressed in Adenocarcinoma of the Lung)/4.1B can regulate substrate methylation via protein arginine N-methyltransferase 3 (PRMT3) and protein arginine methyltransferase 5 (PRMT5), while hypomethylation of cellular proteins, in turn, increases DAL-1/4.1B protein levels [58]. Thus, DAL-1/4.1B tends to act synergistically with post-translational protein methylation to regulate the caspase 8-dependent pathway to induce apoptosis in tumor cells, which in turn regulates the progression of breast cancer. Meningioma is the second-most common brain tumor in adults [59]. In addition, in benign brain tumors, deletion of protein 4.1B expression is one of the most frequently observed changes, i.e., deletion of protein white 4.1B is one of the early events in meningioma [60]. Gerber et al. demonstrated that protein 4.1B in meningioma could activate the Jun N-terminal kinase (JNK) by activating mixed-lineage kinase 3 (MLK3), which sequentially activates Src, rac1, and JNK. This activation is often dependent on the U2 structural domain of protein 4.1B. In contrast, inhibition of rac1 or JNK activation abolishes the regulation of cell growth and cyclin A by protein 4.1B, demonstrating that protein 4.1B is a vital cell growth regulator in meningioma cells [60]. In prostate cancer, protein 4.1B inhibits tumor invasion [61]. Wong et al. showed that protein 4.1B expression was reduced in highly metastatic tumors and that downregulation of protein 4.1B increased the propensity of tumor cells to metastasis [42]. Renal cell carcinoma is a common urological malignancy, with patients with renal clear cell carcinoma accounting for approximately 75% of these diagnoses [62]. Nagata et al. showed that cell adhesion molecule 4 (CADM4) is frequently associated with DAL-1/4.1B in human proximal tubules and that disruption of the CADM4-4.1B cascade is frequently seen in patients with renal cell carcinoma, with the two often acting in concert. Animal studies showed that the mean size of tumors with CADM4 or protein DAL-1/4.1B deficiency was significantly larger than those expressing both CADM4 and 4.1B, explaining that their cascade response may affect intercellular adhesion, i.e., CADM4-4.1B can influence tumor invasion [63]. In Cavanna’s study, the deletion of protein 4.1B resulted in significantly enhanced motility of non-metastatic sarcoma cells, which migrated at a significantly reduced rate if protein 4.1B was expressed exogenously. Protein 4.1B downregulation may also promote the ability of E-26-related genes to mediate the invasion and metastasis of prostate cancer cells. In esophageal squamous carcinoma, protein 4.1B/DAL-1 could inhibit the migratory and invasive effects of esophageal squamous carcinoma cells by inhibiting matrix metalloproteinase matrix metallopeptidase 2 (MMP2) and matrix metallopeptidase 9 (MMP9) [64].

## 6. Clinical Application of Cytoskeleton Protein 4.1 in the Tumor Microenvironment

Biomarkers usually refer to indicators that can be objectively measured and evaluated and respond to physiological or pathological processes [65,66]. In the field of oncology, biomarkers are usually substances produced by tumor cells or other components that can reflect the presence and changes of tumor cells in the body [67]. Furthermore, they should have a certain specificity and sensitivity, as well as reproducibility. From the above discussion, we know that the downregulation or deletion of cytoskeleton protein 4.1 expression is often associated with the occurrence and development of cancer and is a potential biomarker with the above-mentioned properties. Moreover, cytoskeleton protein 4.1 has the potential to be used as a therapeutic target for cancer. In the following, we review the clinical applications of cytoskeleton protein 4.1.

### 6.1. Cytoskeleton Protein 4.1 May Be a Biomarker in Cancer

The failure to detect cancer in time is one of the primary reasons for the high mortality rate of cancer patients, and many patients are already in the middle and late stages of the disease when detected, which seriously affects their subsequent treatment [68]. One of the most important ways to solve this problem is through biomarkers [69]. With the advancement of cancer research, biomarkers have received unprecedented attention in oncology [70]. The search for new and more effective biomarkers is still a vital task. Cytoskeleton protein 4.1, an important class of membrane skeletal proteins, was initially identified in the erythrocyte membrane. As research progressed, its association with immune cells and tumor cells in the tumor microenvironment was also uncovered and its essential role in tumorigenesis and progression was revealed. At the protein level, cytoskeleton protein 4.1 is a direct performer of human life activities and can directly influence biological processes such as cell proliferation. During tumor development, differential expression of cytoskeleton protein 4.1 is often stably detected [71]. Downregulation or even deletion of cytoskeleton protein 4.1 members can be detected in a range of cancers such as meningioma, non-small cell lung cancer, and epithelial ovarian cancer, and the expression of cytoskeleton protein 4.1 members often correlates with the malignancy of the tumor, i.e., patients with downregulated expression are more likely to turn malignant, demonstrating the significant potential of cytoskeleton protein 4.1 as a tumor marker [72]. Currently, for the study of the protein 4.1 families, we can also rely on proteomics, through which we can obtain a complete protein profile of cytoskeleton protein 4.1 for a more detailed analysis of the tumorigenesis and progression process, which will then be of great importance for the diagnosis of cancer patients.

### 6.2. Cytoskeleton Protein 4.1 May Be a Therapeutic Target in Cancer

The main cancer treatments are surgery, radiotherapy, and chemotherapy, but as research into cancer progresses and technology improves, immunotherapy has entered the spotlight with relatively few side effects and a wide range of benefits [73]. Conventional therapy aims to target the tumor itself, but tumors live in a complex tumor microenvironment [74]. If the effect of the tumor microenvironment on the tumor cannot be regulated, the therapeutic effect is significantly reduced, and improvements in immune function will benefit tumor treatment [75]. Cytoskeleton protein 4.1 is associated with both immune cells and malignant tumor cells within the tumor microenvironment, and its high expression often gives the immune system an advantage [76]. In non-small cell lung cancer, for example, cytoskeleton protein 4.1 can influence cancer progression by regulating specific metabolic pathways. Its expression directly affects the proliferation, invasion, and migration of tumor cells while enhancing the activity of immune cells and giving them an advantage [77]. Secondly, the tumor microenvironment serves as the site of tumor cell genesis and development, so improving the tumor microenvironment will affect the progression of tumor development [78,79]. Cytoskeleton protein 4.1 is a tumor metastasis suppressor, and there are multiple ways to restore or promote cytoskeleton protein 4.1 expression, including targeting partners that interact with cytoskeleton protein 4.1 [46]. The recovery or enhancement of cytoskeleton protein 4.1 expression is beneficial for cancer therapy and is a potential therapeutic target [34,35].

## 7. Conclusions

Immunotherapy is an essential branch of tumor treatment. As science and technology continue to advance, its efficacy is improving, with the tumor microenvironment and immune cells attracting attention because of their importance in tumor immunotherapy. Influencing immunity and cancer development by regulating the tumor microenvironment and associated cells have also become increasingly important. As an introductory class of cytoskeleton protein 4.1s, cytoskeleton protein 4.1 has gradually emerged as an essential player in tumor immunity. Based on these findings, we suggest that cytoskeleton protein 4.1 can act as a diagnostic and prognostic marker for tumors within the tumor microenvironment and potentially act as a therapeutic target for tumors. An in-depth understanding of the role and mechanism of cytoskeleton protein 4.1 in the tumor microenvironment will provide a new approach and new ideas for future cancer diagnosis and treatment.

## Figures and Tables

**Figure 1 ijms-24-03777-f001:**
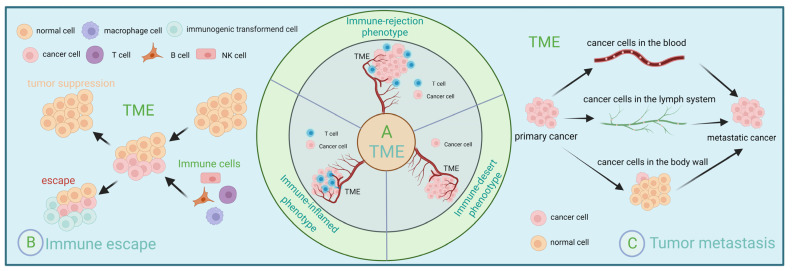
Immunoregulation and tumor microenvironment. (**A**) Depending on the number and spatial distribution of T cells, the tumor microenvironment can be divided into three types: immune inflammatory, immune rejection, and immune desert. (**B**) Schematic representation of immune evasion within the tumor microenvironment. (**C**) Three ways of tumor metastasis: lymphatic metastasis, hematogenous metastasis, and implantation metastasis.

**Figure 2 ijms-24-03777-f002:**
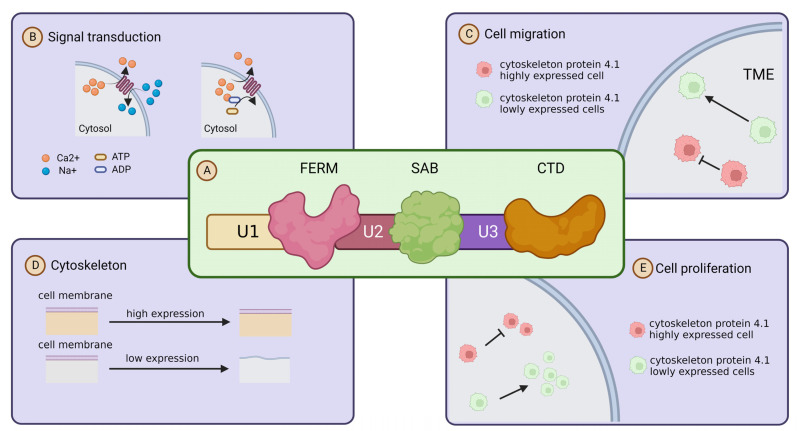
Structure and function of cytoskeleton protein 4.1. (**A**) Schematic diagram of the primary structure of cytoskeleton protein 4.1: Cytoskeleton protein 4.1 family members are composed of three conserved domains (FERM, SABD, CTD) and three unique domains (U1, U2, U3). Their location structure diagram is shown in the illustration. (**B**) Cytoskeleton protein 4.1 and signaling transduction: Cytoskeleton protein 4.1 affects signal transduction by influencing ion transport. (**C**) Cytoskeleton protein 4.1 and cell migration: high expression of cytoskeleton protein 4.1 promotes cell migration. (**D**) Cytoskeleton protein 4.1 and cell membrane stability: high expression of cytoskeleton protein 4.1 maintains the stability of the cell membrane. (**E**) cytoskeleton protein 4.1 and proliferation: high expression of cytoskeleton protein 4.1 inhibits the cell proliferation.

**Figure 3 ijms-24-03777-f003:**
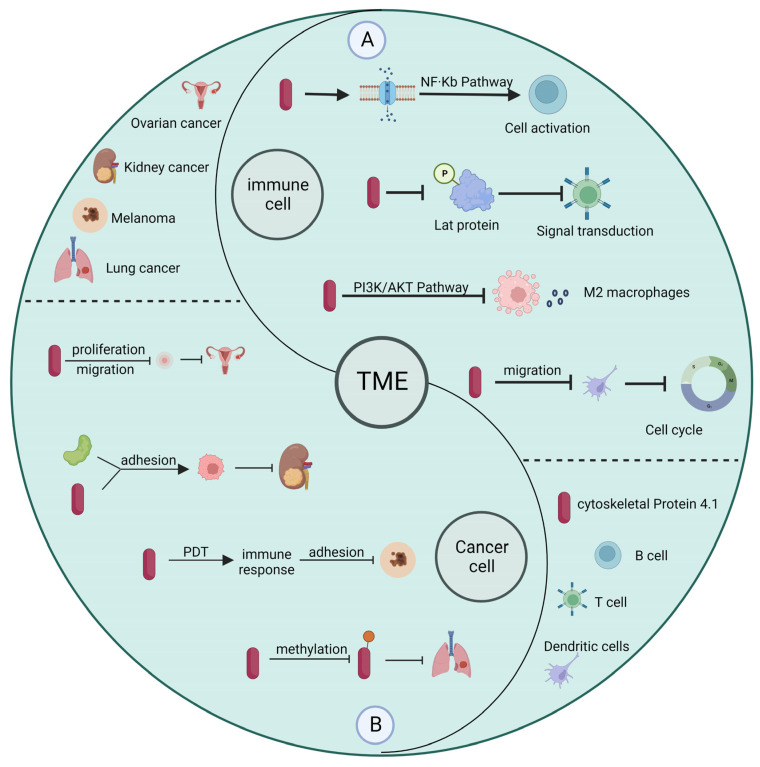
Relationship between cytoskeleton protein 4.1 in tumor microenvironment and immune cells and tumor cells. (**A**) In immune cells, cytoskeleton protein 4.1 affects cell activation, signal transduction, and cell cycle, among others. (**B**) In cancer cells, cytoskeleton protein 4.1 influences the progression of cancer by affecting the malignant characteristics of tumor cells.

**Figure 4 ijms-24-03777-f004:**
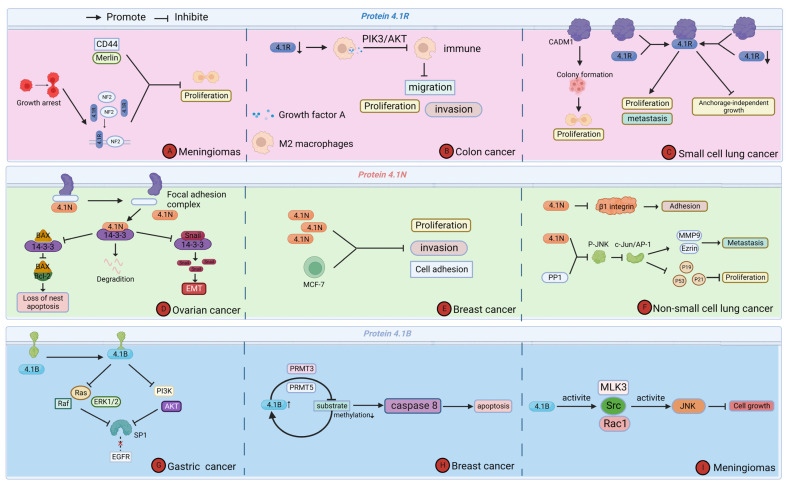
Function of Cytoskeleton Protein 4.1 in tumor microenvironment regulating cancer progression. (**A**) Protein 4.1R affects the proliferation of meningioma. (**B**) Protein 4.1R affects the proliferation, migration, and invasion of colon cancer. (**C**) Protein 4.1R affects the proliferation and metastasis of small cell lung cancer. (**D**) Protein 4.1N affects the apoptosis and degradation of ovarian cancer. (**E**) Protein 4.1N affects the proliferation, invasion, and cell adhesion of breast cancer. (**F**) Protein 4.1N affects the proliferation, metastasis, and adhesion of non-small cell lung cancer. (**G**) Protein 4.1B affects the progression of gastric cancer. (**H**) Protein 4.1B affects the apoptosis of breast cancer. (**I**) Protein 4.1B affects the cell growth of meningioma.

## Data Availability

Not applicable.

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
