# Peer review of "The Role of Cytoskeleton Protein 4.1 in Immunotherapy"

_ijms, 2023, doi:10.3390/ijms24043777_

Round 1

Reviewer 1 Report (Previous Reviewer 4)

The authors made an appreciate efforts to improve manuscript quality, nerveless it remains more improvement to do in Terms of figures image quality and legend text

The text in the figures is not very clear (fuzzy), the quality of the image is pixelated specifically in the figure 1 and 4. The legends are incomplete; In the figure 1 the authors highlight some events with numbers in circle with arrows, but they didn't give any explanation text of that not in the paragraph text nor in the legend figure text.

What are the meanings of 1, 2, 3 or 4 in the figure, if they are useful for the understanding of the figure they should be explained if not needed, it should be removed.

The same for the figure 3 as I mentioned before, there is numbers (1 to 8) which are not referenced anywhere in the legend text.

The reader should be able to understand the figures in the review just by reading the caption. this is not the case in this paper. the legends must imperatively be better completed.

Author Response

Reviewer 2 Report (New Reviewer)

The review by Si et al gives a fresh angle within tumor biology and immunology of interest for the oncology community in general but also to the immune oncology community in particular. Overall the review is outlined in a proper manner and the figures are very good and enlightening.

Some comments to further improve the review:

1) line 23: the authors states that immunotherapy has low toxicity and outstanding efficacy. I would tone that down a little as checkpoint blockade and CAR T cells can have rather harsh side effects but of course that may be a less problem than chemotherapy associated adverse reactions. Also, I suggest to say that the efficacy is outstanding in some patients as a high proportion of complete responses and/or long term responses have been noted compared to conventional therapy but there are still room for improvement as many are primary resistant or become resistant also to immunotherapies.

2) line 26: the second sentence states “…many patients, but many patients…” please rephrase this sentence to avoid the repetition and thereby gain better flow reading it.

3) line 27: do the authors mean “suppressive responses” rather than “specific responses”?

4) line 39: I suggest “via multiple” or “via interaction with multiple” rather than “in multiple”.

5) line 41: I suggest “to elucidate” instead of “and elucidate”.

6) line 53: please remove “which is home to tumor cell growth” that seems redundant.

7) line 55: the authors states that the tumor microenvironment responds to immunotherapy by creating a suppressive microenvironment but the environment is rather suppressive also before immunotherapy even if further mechanisms can be triggered by immunotherapy. Perhaps rephrase this section?

8) The figure legends would benefit being more descriptive to match the very detailed and very good figures.

9) line 77/78: this sentence is repeating migration as mechanism twice.

10) line 86: I suggest adding chemotherapy to the list of conventional cancer therapeutics.

11) line 93: please capitalize “b” in B cell.

12) line 102: VEGFA is abbreviated without explanation but later the abbreviation is not used (line 109, 140).

13) line 102, VEGF also affects angiogenesis.

14) line 159, 162: EOC and EMT are abbreviated without description.

15) section 6.2: Cytoskeleton protein 4.1 is suggested to be a therapeutic target. Please elaborate a little on possible ways? I assume it should be methods to enhance its expression?

Author Response

Reviewer 3 Report (New Reviewer)

The manuscript by Ch. Si et al. reviews the cytoskeletal protein 4.1 and its role in cancer. The manuscript needs significant improvement before it can be accepted for publication in IJMS:

1)      Please revise the manuscript for clarity, especially avoiding long sentences as the one in lines 158-164:

“Wang demonstrated that protein 4.1N expression caused a significant decrease in the expression of protein 4.1N by using two EOC cell lines, SKOV3, which endogenously expresses 4.1N, and A2780, which lacks 4.1N expression, and constructing A2780 cells with protein 4.1N overexpression (referred to as A2780 OE), i.e., wild-type protein 4.1N-silenced A2780 cells (A2780 WT), which caused down-regulation of EMT markers such as N-calmodulin expression, inhibited migration and invasion of EOC cells, and promoted the susceptibility of EOC cells to loss-of-nest apoptosis, i.e., protein 4.1N deficiency in ECO patients promoted EMT in the context of the process.”

2)      Please revise all subchapters of chapter 5 for clarity. My recommendation would be to follow the C-C-C (or context-content-conclusion) rule. This rule and other recommendations can be found: https://doi.org/10.1371/journal.pcbi.1005619

3)      Please revise the overall structure of the manuscript. Currently, chapter 5 is followed by subchapters 5.3, 5.2 and 5.3 again.

4)      Please revise the chapter titles for clarity, for example, line 119, “5.Mechanism of Cytoskeleton Protein 4.1 in tumor microenvironment regulating cancer progression”. Should the word “mechanism” be changed to “function”?

5)      Please explain the meaning of abbreviations or avoid them if possible (e.g. “14-3-3” in line 166, CADM1 in line 146, etc.).

6)      Lines 125-127: “Given the paucity of studies on 4.1G proteins in cancer, we only review the mechanisms by which other members of cytoskeletal protein 4.1 affect cancer in the tumor microenvironment (Figure 4).” Please at least cite some good references about cytoskeletal protein 4.1G function in cancer.

7)      Please carefully check the citation of references – e.g., lines 47-50: “Depending on the number and spatial distribution of T cells, the tumor microenvironment can be divided into three phenotypes: immune-inflammatory, immune-rejection and immune-desert, and thus immunotherapy has different effects on different phenotypes[7]”. However, while reading the reference [7] (Si et al. 2021) I could not find the cited information about the three mentioned phenotypes.

8)      Please pay attention to the citation: e.g., lines 130-133: “A study by Victoria et al. showed that protein 4.1R is frequently expressed at reduced levels or even absent in meningioma patient tissues compared to normal subjects, and the same results were found in two meningioma cell lines IOMM-Lee, CH157-MN[20]”. If “Victoria et al.” is the reference [20], then it should be cited as “Robb et al.” since Victoria is the first name, while Robb is the last.

9)      Inconsistent spelling: is it “cytoskeletal protein 4.1” or “cytoskeleton protein 4.1”?

10)  The figures are prepared using a too-small font to be readable, especially in printed format. However, even viewing pdf file does not help much since the resolution of the images is relatively low.

11)  The figures should be described in detail in the manuscript text or figure legends. For example, what is marked as different color objects in figure 2 in panels C, D, E? What do U1, U2, and U3 mean in figure 2 A? Please check all figures to be understandable to the readers.

Round 2

Reviewer 1 Report (Previous Reviewer 4)

The authors made an appreciable effort on the edition of this manuscript. They seriously improve the quality of the figures, adding a legend text.

The review is now fluent more digest and easy to read.

Author Response

Thank you very much for your recognition of our work. We are also very grateful for your valuable suggestions, which have greatly improved the quality of this manuscript.

Reviewer 3 Report (New Reviewer)

The authors have significantly improved the manuscript's overall structure, readability, and especially figure quality and description.

Author Response

Thank you very much for your recognition of our work. We are also very grateful for your valuable suggestions, which have greatly improved the quality of this manuscript.

This manuscript is a resubmission of an earlier submission. The following is a list of the peer review reports and author responses from that submission.

Round 1

Reviewer 1 Report

Many statements about the protein are made with little or no evidence presented to back them up. For example, why is this protein referred to as a cytoskeletal protein?   It is not suggested to belong to one of the currently accepted classes of cytoskeletal protein (actin, microtubules and intermediate filaments). Do the authors propose that it does belong to one of these classes or do they propose that it belongs to a new class of cytoskeletal protein. As a review article introducing the protein to the field of cytoskeletal research, the accepted terminology should be used to allow the work to be understood by the field at large. As it is it is not clear what type of protein this is or what class of protein it belongs to.

Page 2 final sentence of paragraph 2:  meaning unclear. what does the timing refer to? as soon as possible after what step? what is the balance that is referred to? balance between what?

Page 2, paragraph 3: 'linker of the transmembrane cytoskeleton...' unclear. what is the transmembrane cytoskeleton?

Page 2, paragraph 3: 'membrane backbone protein 4.1'. unclear. Is this the same as cytoskeletal protein 4.1?

Page 3, line 1: 'the third of which is specific immunity'.  why is this 3rd? what are the other 2?

Page 3 'recognised oncoprotein'. is it a recognised oncoprotein? THis has not been mentioned previously in this review. What is the evidence for this?

Page 3: 'many studies have shown'. what studies? cite the papers.

Reviewer 2 Report

I would like to present my positive opinion about the reviewed work.  The aim of this study was to Cytoskeletal protein 4.1 affects immunotherapy by regulating the balance of the tumor microenvironment.

In the manuscript, the authors examine very important, and interesting issues related to cytoskeletal protein 4.1 plays a vital role in cancer as a tumor suppressor. In the manuscript, the authors review the latest literature related to the analyzed issue.

The topic has been well discussed with the current literature (most of the works from the last 5 years). Additionally, the attached figures facilitate the assimilation of the issue.

I propose to adapt the manuscript appearance to the requirements of the journal, add figures in the text not at the end of the manuscript and review the reference as required by the journal. After this minor changes, I propose to publish it in the International Journal of Molecular Sciences in Section Molecular Biology.

Reviewer 3 Report

The authors present a review of physiological roles of cytoskeletal protein 4.1 and its subtypes, along with their implications in tumor formation and progression. 

To the best of my knowledge and after reviewing the available literature, there have not been any published reviews covering this specific topic. All key points of the review are highlighted in the abstract.

Although the authors invested a noteworthy effort into studying this topic in detail, I have numerous issues with the manuscript, which I will try to elaborate in a stepwise manner.

1. The Introduction section starts with a very broad and general description of immunotherapy, followed directly by discussing the role and structure of cytoskeletal protein 4.1 and then briefly describing the interplay between 

protein 4.1 and the immune system. 

The main issues with the Introduction section are:

  1. The English style of the Introduction is very atypical for scientific literature - sentences like “Cancer has always been a massive challenge for human health and safety, which has led to the innovation and development of 20 new treatment technologies”, “In the face of non-self components, the body can rely on its immune function to maintain its health and defend itself against 36 infection or other unfriendly states”, “As the home base for tumor cell growth and development, the tumor microenvironment acts like 37 an area of the body occupied by the enemy, while protein 4.1 plays the role of a friendly force, inhibiting the process of cancer 38 development and progression.” are too colloquial and do not belong to a scientific journal.
  2. The introductory part of the article should preferably be written in a “funnel” manner, meaning it should start from a broad perspective and narrow down to the specifics of the article. In the case of this manuscript, the initial paragraph is too broad and can be ommited. Instead, a paragraph describing cytoskeletal proteins in general, would be much more fitting (for example, see Ong, M.S.; Deng, S.; Halim, C.E.; Cai, W.; Tan, T.Z.; Huang, R.Y.-J.; Sethi, G.; Hooi, S.C.; Kumar, A.P.; Yap, C.T. Cytoskeletal Proteins in Cancer and Intracellular Stress: A Therapeutic Perspective. Cancers 2020, 12, 238. https://doi.org/10.3390/cancers12010238).

2. The authors then continue with the section 2. Immunoregulation and tumor microenvironment. Lines 45-52 can be omitted, since the concepts described here are too elementary to be included. 

3. This section is followed by “3. Cytoskeletal protein 4.1.” The first issue with this section, which is continued throughout the manuscript is the use of this open compound word “cytoskeletal protein 4.1”, whereas in the literature, the term used most commonly is Protein 4.1, without the adjective “cytoskeletal”. The sentence “Proteins are the direct performers of life activities and the vast majority of biological life activities need to be performed by proteins.“ is of inadequate language style. The sentence “In conclusion, cytoskeletal protein 4.1 can regulate the homeostasis of the tumor  microenvironment, enhancing the function of immune cells on the one hand and inhibiting various malignant features of tumor cells on the other (Figure2).”, along with the Figure 2 associated, lacks references. The figure includes some typing errors such as “inhibite” instead of “inhibit” and “enpression” instead of “expression”.

4. The Role of Cytoskeleton Protein 4.1 in immunoregulation and cancer - this section aims to discuss the matter mentioned in its title. The statements provided herein, such as “In addition to radiotherapy, which is still the treatment for cancer, and to which most patients are resistant, immunotherapy is gradually entering the public eye with its remarkable therapeutic effects” the authors mention radiotherapy as a modality in cancer treatment, but do so in a rather superficial and vague manner - to which patients are they referring to? Cancer patients in general? There are some cancers which are highly radiosensitive, others are radioresistant and some fall in between - it is unclear why such inaccurate statements are included in this manuscript. 

Lines 112-154 provide numerous specific and rather clear examples of interplay between protein 4.1 and immune cells - this part is scientifically much more sound, is backed by references and is relatively easy to follow. I warmly encourage the authors to build on this section, in order to improve the rest of the manuscript.

5.Cytoskeleton protein 4.1 in tumor microenvironment - this section discusses its topic very superficially, again without specific pathophysiological data which would be appreciated - this is included in the following sections, therefore making section 5 of questionable importance for the manuscript.

6.Mechanism of Cytoskeleton Protein 4.1 in tumor microenvironment regulating cancer progression - in this sections the authors discuss specific roles of protein 4.1 R, 4.1 N and 4.1 B as members of the tumor microenvironment. The sentence “Cytoskeletal protein 4.1 is an important tumor suppressor molecule that is downregulated or absent in various cancers,  including prostate cancer and meningioma” is misleading - meningiomas are termed “cancers”, i.e. malignant tumors, although more than 90% of them are grade 1 - if the authors are implying higher grade meningiomas have this downregulation, they should state this explicitly. The following sections regarding specific protein 4.1 types discuss numerous examples of the involvement of protein 4.1 in tumor formation and progression, and is adequately backed by references. 

7. The section “Clinical application of Cytoskeleton Protein 4.1 in tumor microenvironment“ states that “its association with immune cells and tumor cells in the tumor microenvironment was also uncovered and revealed its essential role in tumorigenesis and progression” with the reference of Greten&Grivennikov’s 2019. Immunity paper, but upon checking, there is no mention of protein 4.1 in this paper, making this reference faulty.

8. The section on future perspectives and limitations is mostly speculative, without any novel information. 

All in all, I would like to encourage the authors to perform a substantial revision of the manuscript - by appointing a native English speaker to help during the preparation of the manuscript, and by reading and analyzing the style and idea of scientific writing from the articles published in International Journal of Molecular Sciences, in order to make this manuscript publishable. 

Reviewer 4 Report

This review needs more work, it’s still in draft form, and needs to be more thorough in some parts. We see a serious willingness of the authors to cover the subject, but some parts are only skimmed over. Lots of missing or mismatched reference errors.

In the paragraph (4) title The role of Cytoskeleton Protein 4.1 in immunoregulation and cancer; line 102, the authors claimed that “the radiotherapy is the only treatment of cancer” but nowadays is not the case it’s systematically associated with other treatments, immunotherapy, Corticotherapy and so… it’s really rare nowadays that cancer patients is treated with only radiotherapy!

In the same paragraph 4., line109-110, authors set ideas as questions, it’s better to discuss and details the papers study then just setting up questions, and there is missing references.

The authors talk about the role of B cell in immune responses, but they didn’t really discuss the role of the Cytoskeleton Protein 4.1 in B cells function.

Be careful, somme references are not corresponding to the text.

In the Paragraph5 the role of Cytoskeleton Protein 4.1 in the microenvironment in immune cells, it’s already discussed a little bit in the paragraph before and seems to be repetitions.

The text in the figures is not very clear (fuzzy). The legends are incomplete, in figure 3 we find numbers (1 to 8) which are not referenced in the legend text.